# STOCHASTIC SPARSE SAMPLING: A FRAMEWORK FOR VARIABLE-LENGTH MEDICAL TIME SERIES CLASSIFICATION

## ABSTRACT

While the majority of time series classification research has focused on modeling fixed-length sequences, variable-length time series classification (VTSC) remains critical in healthcare, where sequence length may vary among patients and events. To address this challenge, we propose **S**tochastic **S**parse **S**ampling (SSS), a novel VTSC framework developed for medical time series. SSS manages variable-length sequences by sparsely sampling fixed windows to compute local predictions, which are then aggregated and calibrated to form a global prediction. We apply SSS to the task of seizure onset zone (SOZ) localization, a critical VTSC problem requiring identification of seizure-inducing brain regions from variable-length electrophysiological time series. We evaluate our method on the Epilepsy iEEG Multicenter Dataset, a heterogeneous collection of intracranial electroencephalography (iEEG) recordings obtained from four independent medical centers. SSS demonstrates superior performance compared to state-of-the-art (SOTA) baselines across most medical centers, and superior performance on all out-of-distribution (OOD) unseen medical centers. Additionally, SSS naturally provides post-hoc insights into local signal characteristics related to the SOZ, by visualizing temporally averaged local predictions throughout the signal.

## 1 INTRODUCTION

Artificial intelligence (AI) in medicine has received significant attention in recent years, with various applications to clinical diagnosis and treatment planning (Rajpurkar et al., 2022). Despite its advancements, the actual integration into everyday clinical practice remains limited, with much of it attributed to the challenges of handling the complexity and variability in medical data. One particularly challenging aspect of this variability lies in the nature of medical time series data. Variable-length time series are prevalent throughout many areas of healthcare, including heart rate monitoring, blood glucose measurements, and electrophysiological recordings where sequence length can vary dependent on the recording or length of an event (Agliari et al., 2020; Deutsch et al., 1994; Walther et al., 2023). Yet, the majority of time series classification (TSC) literature focuses solely on methods that process fixed-length sequences (Ismail Fawaz et al., 2019; Mohammadi Foumani et al., 2024).

At the same time, healthcare applications require greater interpretability from modern time series methods to expand their applicability in critical domains and accelerate clinical adoption (Amann et al., 2020). This interpretability is especially crucial in contexts where the relationship between pathology and signal characteristics is not well understood, as it can provide valuable insights for both clinicians and scientists. Recent studies in time series classification (TSC) have explored the explainability of specific signal segments, as opposed to full-signal analysis, which proves particularly useful for uncovering important characteristics

such as motifs, anomalies, or frequency patterns (Early et al., 2024; Crabbé & Van Der Schaar, 2021; Huang et al., 2024). However, there still remains a significant need for models with built-in interpretability in medical applications. Such methods would alleviaate the burden of implementing both a base model and a specialized interpretability method—which may require more domain expertise—and may more effectively facilitate clinical adoption.

The need for variable-length time series classification (VSTC) methods with built-in interpretability is particularly relevant in seizure onset zone (SOZ) localization—the task of identifying brain regions from which seizures originate—as effective treatment requires analysis of variable-length signals (Balaji & Parhi, 2022). The World Health Organization (WHO) reports epilepsy affects over 50 million people globally, establishing it as one of the most common yet poorly understood neurological disorders (Organization et al., 2019; Stafstrom & Carmant, 2015). Additionally, one-third of patients do not respond to antiepileptic drugs, making surgery the last resort and accurate SOZ localization essential for effectively planning the operation. The process of SOZ identification involves a two-step procedure: initial implantation of electrodes in areas suspected to contain the SOZ, followed by recording and visual analysis of intracranial electroencephalography (iEEG) signals by medical experts. The task of SOZ localization reduces to classifying individual electrode recordings, representing different regions within the brain. Effective localization of the SOZ is challenging due to the absence of clinically validated biological markers and the variable-length nature of iEEG signals—consequently, surgical success rates range from 30% to 70% (Löscher et al., 2020; Li et al., 2021).

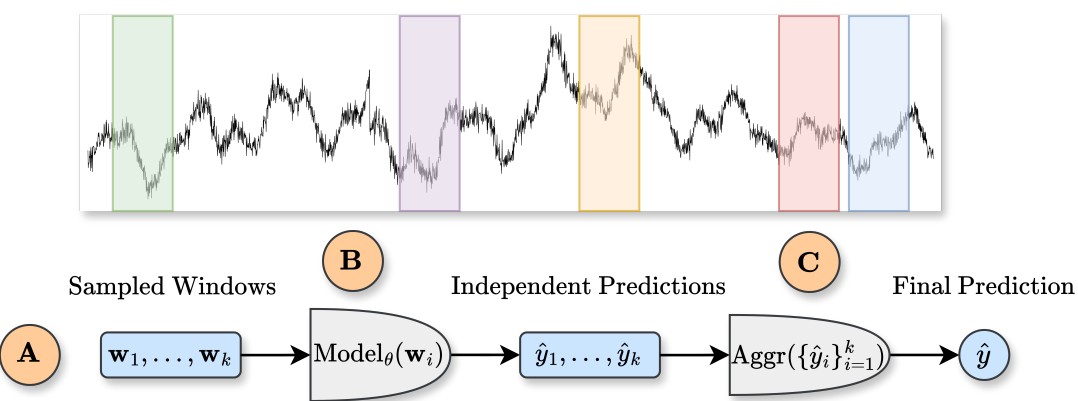

Figure 1: An overview of Stochastic Sparse Sampling (SSS) training procedure. (**A**) For a given time series, we sample windows of fixed-length at random throughout the signal. (**B**) Each window is processed independently by a local model with parameters $\theta$, outputting the local predictions $\hat{y}_1, \ldots, \hat{y}_k$. (**C**) Local predictions are then fed through an aggregation function to form the final prediction $\hat{y}$.

**Contributions**. While our work primarily focuses on VTSC, we also evaluate our method's performance on OOD data and explore its potential for providing local explanations. To this end, we propose **S**tochastic **S**parse **S**ampling (SSS) a novel framework for VTSC developed for medical time series. The main contributions of our paper are listed as follows:

- **Robustness to long and variable-length sequences**. SSS samples fixed-length windows, and processes them independently through a single model. This prevents context overload in long sequences seen in infinite-context methods, and does not utilize padding, truncation, or interpolation

required by finite-context methods. By relying on a single local model, SSS utilizes far fewer parameters compared to finite-context methods that traditionally ingest the entire signal, which significantly reduces computational cost during training and the risk of overfitting over long sequences.

- **Generalization to unseen patient populations.** SSS demonstrates strong performance on out-of-distribution (OOD) data from unseen medical centers. When trained on data from one or more medical centers and evaluated on a completely new center with a different patient population, SSS outperforms all baselines in our comparisons. This result suggests SSS's potential as a foundation model for TSC, opening new avenues for research and clinical applications.

- **Explainability through local predictions**: Our method enhances model interpretability by directly tying each output—a probability score for each window—to the overall prediction. This capability is crucial in critical clinical settings, such as SOZ localization, which traditionally relies on visual analysis. Given the significant risks associated with brain region removal, any proposal should be designed to integrate within clinical workflows. Moreover, in the absence of universally recognized biological markers for epilepsy, SSS offers the potential to further our understanding the SOZ and to identify novel markers.

- **Compatibility with modern and classical backbones.** SSS integrates with any time series backbone. This ensures that our approach leverages well-established frameworks now and into the future, allowing for adaptability across a diverse array of contexts.

## 2 RELATED WORK

TSC methodologies can be broadly categorized into finite-context methods, which operate on fixed-length input segments, and infinite-context methods, which handle variable-length sequences without being restricted to a predetermined window size. For a formal treatment, please see Appendix A.5.

**Finite-context methods**. Finite-context methods are among the most commonly used approaches for TSC. Transformer-based models have gained significant attention, with variations such as sparse attention, series decomposition, and patching techniques (Vaswani et al., 2017; Kitaev et al., 2020; Zhou et al., 2021; Wu et al., 2021; Zhou et al., 2022; Liu et al., 2022; Nie et al., 2023; Liu et al., 2024). Several temporal convolutional networks (TCNs) have also been proposed, to capture temporal dependencies through dilated convolutions and Inception-like architectures (Lai et al., 2018; Bai et al., 2018; Ismail Fawaz et al., 2020; Wu et al., 2022; Luo & Wang, 2024). Recently, multilayer perceptrons (MLPs) and simple linear models have demonstrated competitive performance as well (Chen et al., 2023; Zeng et al., 2023). Despite the significant rise of finite-context methods, these methods are inherently limited in their ability to handle variable-length sequences, and will require the use of either padding, truncation, or interpolation for VTSC. Furthermore, as the sequence length increases, so does the number of model parameters, which leads to not only greater computational cost but an increased risk of overfitting.

**Infinite-context methods**. The recurrent neural network (RNN) family includes several models capable of ingesting variable-length time series (Rumelhart et al., 1986). Long-short term memory (LSTM) networks introduce memory cells and gating mechanisms to better handle long-term dependencies (Hochreiter & Schmidhuber, 1997). Gated recurrent units (GRUs) simplify the LSTM architecture while maintaining similar performance (Bahdanau et al., 2014). State space models (SSMs) have gained recent attention, with approaches such as S4 introducing structured parameterization to enable efficient computation over long sequences, while still attempting to capture long-range dependencies (Gu et al., 2021). Building on this, Mamba introduces a selective SSM that adapts to input dynamics, further improving processing of

long-range dependencies in time series data while while maintaining linear time complexity with respect to sequence length (Gu & Dao, 2023). Despite these advancements, RNNs and SSMs can still struggle with retaining information in extremely long sequences and may be prone to vanishing or exploding gradients (Salehinejad et al., 2017). ROCKET offers an alternative, using random convolutional kernels to convert input into a fixed-length representation for VTSC, but at the cost of interpretability and potentially limited model expressivity (Dempster et al., 2020).

**SOZ localization methods**. Several recent proposals have been tailored specifically to SOZ localization. Functional connectivity graphs compute patient-specific channel metrics to capture brain connectivity patterns (Grattarola et al., 2022; Fang et al., 2024), offering insights into functional relationships associated with seizures. However, their reliance on intra-patient dynamics makes them unsuitable for a single model that can generalize across multi-patient, heterogeneous datasets. Alternatively, electrical stimulation methods that use intracranial electrodes (Johnson et al., 2022; Yang et al., 2024) can enhance localization accuracy through induced responses analyzed by TCNs and logistic regression models. Yet, these approaches require both fixed-length windows and the use of active stimulation. For our purpose of building a general model for SOZ localization, which can be applied to multiple patients (with a potentially varying number of channels) without electrical stimulation, we do not consider such approaches in our study.

## 3 METHOD

### 3.1 VARIABLE-LENGTH TIME SERIES CLASSIFICATION

Consider a collection of time series $X = \left\{ (\mathbf{x}_t^{(1)})_{t=1}^{T_1}, \ldots, (\mathbf{x}_t^{(n)})_{t=1}^{T_n} \right\}$ with labels $Y = \{y^{(1)}, \ldots, y^{(n)}\}$, where each series $i$ has sequence length $T_i \in \mathbb{N}$, and for each time point $t$, the vector $\mathbf{x}_t^{(i)} \in \mathbb{R}^{M_i}$ has $M_i \in \mathbb{N}$ channels. The goal of VTSC is to learn a classifier $f_\theta$ which maps each series $(\mathbf{x}_t^{(i)})_{t=1}^{T_i}$ to its corresponding class in $\{1, \ldots, K\}$ for $K \in \mathbb{N}$ classes. We require that $f_\theta$ can handle sequences of any length—that is, it has infinite context—since we assume that each $T_i$ can be arbitrarily large at inference time. Otherwise, we must adjust a finite-context classifier using padding, truncation, or interpolation.

### 3.2 STOCHASTIC SPARSE SAMPLING

#### 3.2.1 SPARSE TRAINING

Figure 1 provides an overview of SSS at train time. During each training epoch, SSS performs a sampling procedure without replacement to create each batch. Fix $L \in \mathbb{N}$ as the window size and let $\mathcal{W}$ be the collection of all windows with size $L$ from all time series in $X$. Within any batch, a window is drawn from $\mathcal{W}$, where the probability of it originating from series $i$ is set to $p_i \approx T_i / \left( \sum_{j=1}^n T_j \right)$ for every $i$. More formally, for each $i$, let $N_i$ be the random variable representing the number of windows from series $i$ in a batch of size $B$. Then $N_i \sim \text{Binomial}(B, p_i)$, and consequently $\mathbb{E}[N_i] = Bp_i$. This proportional sampling ensures fair representation of each series based on its length, allowing longer sequences—which contain more information—to contribute more samples. By sampling only a subset of windows, SSS introduces sparsity into the training process, reducing computational cost found in finite-context methods, and the likelihood of context overload in infinite-context methods. Also note that by sampling with replacement, the model sees each window exactly once during a single training epoch.

After sampling a batch of windows $\mathcal{W}_0 = \{\mathbf{w}_1, \ldots, \mathbf{w}_B\}$, each $\mathbf{w}_b \in \mathcal{W}_0$ is processed independently by a local model $f_\theta$ to obtain a local prediction $\hat{y}_b = f_\theta(\mathbf{w}_b) \in [0, 1]^K$, representing our probability distribution over $K \in \mathbb{N}$ classes. The choice of $f_\theta$ can be any time series backbone, in our experiments we select PatchTST (Nie et al., 2023). For each time series $1 \le i \le n$, denote:

$$\mathcal{W}_i = \{\mathbf{w} \in \mathcal{W}_0 \mid \mathbf{w} \text{ is from series } i\}, \tag{1}$$

as the collection of windows in the batch originating from series $i$, and let:

$$\mathcal{Y}_i = \{f_\theta(\mathbf{w}) \mid \mathbf{w} \in \mathcal{W}_i\}, \tag{2}$$

be the set of multiset of window probabilities. To obtain the global prediction for time series $i$, we aggregate window probabilities from all examples originating from it, given by:

$$\hat{y}^{(i)} = \mathrm{Aggr}(\mathcal{Y}_i) = \sum_{\hat{y} \in \mathcal{Y}_i} \alpha(\hat{y})\hat{y}, \tag{3}$$

where each $\alpha(\hat{y}) \in [0,1]$ represents the weight of window probability $\hat{y}$ to the final output, satisfying $\sum_{\hat{y} \in \mathcal{Y}_i} \alpha(\hat{y}) = 1$; that is, $\mathrm{Aggr}(\cdot)$ produces a convex combination over $\hat{y}_1, \ldots, \hat{y}_n$. In our experiments, we use mean aggregation, i.e., $\alpha(\hat{y}) = \frac{1}{|\mathcal{Y}_i|}$ for all $\hat{y} \in \mathcal{Y}_i$, due to its simplicity and effectiveness for our current objectives. This formulation guarantees that $\hat{y}^{(i)}$ remains a valid probability distribution over $K$ classes (proof in Appendix A.3), and allows for potential of non-uniform aggregation functions, enabling weighting of window predictions based on factors such as prediction uncertainty, or frequency characteristics.

---

**Algorithm 1** SSS Training Algorithm (Single Epoch)

---

**Input:** Time series $X = \{(\mathbf{x}_t^{(1)})_{t=1}^{T_1}, \ldots, (\mathbf{x}_t^{(n)})_{t=1}^{T_n}\}$, labels $Y = \{y^{(1)}, \ldots, y^{(n)}\}$, model $f_\theta$ with parameters $\theta$, batch size $B$
**Output:** Updated model parameters $\theta$
$\mathcal{W} \leftarrow$ Set of all windows from each series in $X$
**while** $\mathcal{W} \neq \emptyset$ **do**
    $\triangleright$ Sample $B$ windows with probability $T_i / (\sum_j T_j)$ from series $i$, for all $i$
    $\mathcal{W}_0 \leftarrow \mathrm{SAMPLE}(\mathcal{W}, B)$
    **for** $i = 1, \ldots, n$ **do**
        $\mathcal{W}_i \leftarrow \{\mathbf{w} \in \mathcal{W}_0 \mid \mathbf{w} \text{ is from series } i\}$                $\triangleright$ Windows from series $i$
        $\mathcal{Y}_j \leftarrow \{f_\theta(\mathbf{w}) \mid \mathbf{w} \in \mathcal{W}_j\}$          $\triangleright$ Window probabilities for series $i$
        $\hat{y}^{(i)} \leftarrow \mathrm{AGGREGATE}(\mathcal{Y}_j)$             $\triangleright$ Final probability for series $i$
    $\mathcal{L}_{\text{batch}} \leftarrow \frac{1}{n} \sum_{i=1}^n \mathcal{L}(\hat{y}^{(i)}, y^{(i)})$          $\triangleright$ Loss over the batch
    $\theta \leftarrow \mathrm{UPDATE}(\theta, \mathcal{L}_{\text{batch}})$          $\triangleright$ Update local model parameters
    $\mathcal{W} \leftarrow \mathcal{W} \setminus \mathcal{W}_0$          $\triangleright$ Remove sampled windows from the pool
**return** $\theta$

---

### 3.2.2 INFERENCE

To the derive the prediction for a time series $(\mathbf{x}_t^{(i)})_{t=1}^{T_i}$ at inference time, we utilize all windows from the selected time series, to form its final prediction $\hat{y}^{(i)}$. Let $\mathcal{W}_i$ be the collection of all windows from series $i$. We pass each window through the local model to obtain the multiset of window probabilities $\mathcal{Y}_i$ as shown in Equation (2). Before the aggregation step, we utilize a calibrator $g_\phi : [0,1]^K \to [0,1]^K$, which adjusts each individual window probability to reduce the presence of noise, and define:

$$\tilde{\mathcal{Y}}_i = \{g_\phi(\hat{y}) \mid \hat{y} \in \mathcal{Y}_i\}, \tag{4}$$

which is then fed into the final prediction during the aggregation step $\hat{y}^{(i)} = \mathrm{Aggr}(\tilde{\mathcal{Y}}_i)$. By calibrating the window probabilities before aggregation, we correct for biases or misestimations in the predicted probabilities, which occur when the output probabilities of the local models do not accurately reflect the true likelihood of the event. We consider isotonic regression and Venn-Abers predictors for our calibration method. It is important to note, that these calibration techniques do not alter underlying structure of $f_\theta$ and do not utilize input features from the time series; rather, they adjust the output probabilities to mitigate the effect of noise during the aggregation step. For more information regarding calibration methods see Appendix A.4.

# 4 EXPERIMENTS

## 4.1 BASELINES

For our finite-context baselines, we include a variety of modern time series backbones including PatchTST, which uses subwindows as tokens in combination with the traditional Transformer architecture (Nie et al., 2023; Vaswani et al., 2017). TimesNet is a TCN architecture that models both interperiod and intraperiod dynamics by leveraging Fast Fourier Transform (FFT) features to slice the signal into multiple views, which are then processed through inception blocks (Wu et al., 2022; Ismail Fawaz et al., 2020). ModernTCN, is a recently proposed TCN which decouples temporal and channel information processing by using separate DWConv and ConvFFN modules for more efficient representation learning (Luo & Wang, 2024). DLinear is linear neural network, which has shown to outperform several modern Transformer-based architecutres, utilizing traditional seasonal-trend decomposition techniques (Zeng et al., 2023). In our infinite-context baselines, we utilize ROCKET, which applies randomly initialized, fixed convolutional kernels to the input sequence. ROCKET compresses the resulting convolutional outputs to the maximum value and the proportion of positive values (PPV), where these features then fed to a linear classifier (Dempster et al., 2020). We also consider GRUs and an LSTM network, both of which are popular RNN frameworks designed to capture long-term dependencies in sequential data (Bahdanau et al., 2014; Hochreiter & Schmidhuber, 1997). Additionally, use the recent SSM architecture, Mamba, which utilizes selective state updates to enable efficient long-range dependency modeling (Gu & Dao, 2023). Further details regarding configurations and hyperparameter tuning for each baseline can be found in Appendix C.

## 4.2 DATASET

The Epilepsy iEEG Multicenter Dataset[1] consists of iEEG signals with SOZ clinical annotations from four medical centers including the Johns Hopkins Hospital (JHH), the National Institute of Health (NIH), University of Maryland Medical Center (UMMC), and University of Miami Jackson Memorial Hospital (UMH). Since UMH contained only a single patient with clinical SOZ annotations, we did not consider it in our main evaluations; however, we did use UMH within the multicenter evaluation in 1 and the training set for OOD experiments for SOZ localization on unseen medical centers in Table 2. We select the F1 score, Area Under the Receiver Operator Curve (AUC), and accuracy for our evaluation metrics. For summary statistics and information on the dataset see Appendix B.1.

## 4.3 UNIVARIATE VTSC

For each patient iEEG recording, the goal of SOZ localization is to determine the the correct of channels or electrodes which belong the seizure onset zone. This effectively reduces the task to univariate TSC. While several channel-dependent methods have been proposed for SOZ localization (see section 2), we focus primarily on channel-independent solutions for two key reasons: (1) they are more resilient to interpatient variability and are unaffected by factors like channel count and therefore can be applied in multiple hospital settings, and (2) they generalize better to domains beyond electrophysiological data, as they learn local signal characteristics rather than explicitly modeling functional connectivity between electrode sites, which may not be present in other medical time series.

Table 1 summarizes our experimental results for SOZ localization on each individual medical center, along with training and evaluation on all medical centers. Within the multicenter evaluation, SSS outperforms all baselines for each evaluation metric. SSS also shows strong performance for the JHH and NIH centers, with comparable results in the UMMC center. We attribute this difference in performance for UMMC due to the fact that it is the only center where the sampling frequency of patient recordings can differ between patients

---

[1] https://openneuro.org/datasets/ds003029/versions/1.0.7

Table 1: SOZ localization. F1 score, AUC, and Accuracy are reported for each medical center, averaged over 5 seeds. For each center, we train and evaluate a separate model; the first column represents training and evaluation on all centers. **Bolded** values with * and † denote the best and second-best results, respectively.

| Model | All | | | JHH | | | NIH | | | UMMC | | |
|---|---|---|---|---|---|---|---|---|---|---|---|---|
| | F1 | AUC | Acc. | F1 | AUC | Acc. | F1 | AUC | Acc. | F1 | AUC | Acc. |
| **SSS (Ours)** | **0.7629*** | **0.7999*** | **72.35*** | **0.8187*** | **0.8851*** | **81.37*** | **0.6716*** | 0.6853 | **64.22†** | **0.7978†** | **0.8279†** | 76.06 |
| PatchTST (Nie et al., 2023) | **0.7097†** | **0.7852†** | 66.83 | **0.7419†** | **0.8045†** | 71.82 | 0.6402 | **0.7036†** | 62.11 | **0.8015*** | 0.8121 | 77.58 |
| TimesNet (Wu et al., 2022) | 0.6897 | 0.7174 | 65.98 | 0.6891 | 0.8029 | **73.64†** | 0.5950 | 0.6806 | **66.00*** | 0.7821 | 0.8099 | **77.06†** |
| ModernTCN (Luo & Wang, 2024) | 0.6938 | 0.7305 | 68.42 | 0.6710 | 0.7508 | 67.73 | 0.5055 | **0.7220*** | 64.00 | 0.6371 | 0.8203 | 71.76 |
| DLinear (Zeng et al., 2023) | 0.6916 | 0.7044 | 68.41 | 0.6873 | 0.7395 | 66.36 | 0.6055 | 0.6405 | 59.50 | 0.7658 | 0.7729 | 77.05 |
| ROCKET (Dempster et al., 2020) | 0.6847 | 0.7481 | 69.27 | 0.6753 | 0.7752 | 69.09 | **0.6520†** | 0.6546 | 62.63 | 0.7686 | 0.7900 | 74.55 |
| Mamba (Gu & Dao, 2023) | 0.6452 | 0.7134 | 64.39 | 0.6456 | 0.6764 | 62.27 | 0.5974 | 0.6050 | 58.95 | 0.7900 | **0.8424*** | 76.36 |
| GRUs (Bahdanau et al., 2014) | 0.6948 | 0.7340 | 65.85 | 0.6140 | 0.6959 | 63.18 | 0.6171 | 0.6283 | 62.63 | 0.7920 | 0.8211 | **77.27*** |
| LSTM (Hochreiter & Schmidhuber, 1997) | 0.6709 | 0.7144 | 65.43 | 0.6571 | 0.6190 | 59.09 | 0.5657 | 0.5909 | 54.74 | 0.7604 | 0.8060 | 73.64 |

(250-1000 Hz), whereas JHH and NIH both have sampling frequencies of 1000 Hz. We also observe that in general, finite-context perform better on the chosen evaluation metrics in comparison to infinite-context methods, for in-distribution univariate VTSC.

## 4.4 OUT-OF-DISTRIBUTION VTSC

Table 2 reports our results for SOZ localization in the OOD setting. At train time, from the collection of four medical center datasets, we omit one and train on the remaining three. At inference time, we test solely on the omitted medical center to gauge how well each method performs OOD. For iEEG signals from epilepsy patients, inter-patient variability can be significant due to differences in placement of electrodes in the brain, and the inherent heterogeneity of epileptogenic networks across individuals. Thus, even among medical time series, this can be one of the most challenging tasks to perform OOD. SSS outperforms each baseline on each unseen medical center, often by a considerable margin when compared to finite-context methods.

Table 2: Out-of-Distribution SOZ localization. F1 score, AUC, and Accuracy are reported for unseen medical centers, averaged over 5 seeds. For each center, we train on all other centers and evaluate on the selected center. **Bolded** values with * and † denote the best and second-best results, respectively.

| Model | JHH | | | NIH | | | UMMC | | |
|---|---|---|---|---|---|---|---|---|---|
| | F1 | AUC | Acc. | F1 | AUC | Acc. | F1 | AUC | Acc. |
| **SSS (Ours)** | **0.6981*** | **0.6590*** | **57.80*** | **0.6492*** | **0.6092*** | **54.73†** | **0.7243*** | **0.8048*** | **72.42*** |
| PatchTST (Nie et al., 2023) | 0.6175 | 0.5267 | 50.46 | 0.5986 | 0.4829 | 48.17 | 0.5067 | 0.5274 | 57.63 |
| TimesNet (Wu et al., 2022) | 0.5261 | 0.4501 | 47.00 | 0.4461 | 0.4407 | 45.85 | 0.3177 | 0.3108 | 46.14 |
| ModernTCN (Luo & Wang, 2024) | 0.4934 | 0.4970 | 49.54 | 0.4019 | 0.4651 | 48.71 | 0.3804 | 0.4474 | 50.55 |
| DLinear (Zeng et al., 2023) | 0.4205 | 0.4775 | 47.25 | 0.5090 | 0.4945 | 50.54 | 0.5602 | 0.5236 | 56.00 |
| ROCKET (Dempster et al., 2020) | 0.5784 | 0.5777 | **56.71†** | 0.5051 | 0.5522 | 52.91 | 0.5608 | 0.5941 | 58.36 |
| Mamba (Gu & Dao, 2023) | 0.5790 | **0.5835†** | 55.68 | **0.6183†** | **0.5767†** | **55.69*** | 0.5715 | 0.5953 | 55.76 |
| GRUs (Bahdanau et al., 2014) | 0.5779 | 0.4868 | 48.80 | 0.5824 | 0.5588 | 53.66 | **0.6689†** | **0.7645†** | **69.30†** |
| LSTM (Hochreiter & Schmidhuber, 1997) | **0.6362†** | 0.5165 | 50.92 | 0.5774 | 0.5678 | 53.87 | 0.6581 | 0.6616 | 62.36 |

In comparison to Table 1, we observe the opposite trend between finite- and infinite-context methods, whereas infinite-context methods seem to perform better OOD. In contrast, SSS demonstrates robust performance both in distribution (where finite-context methods excel) and OOD (where infinite-context methods excel).

## 4.5 QUALITATIVE VISUALIZATION

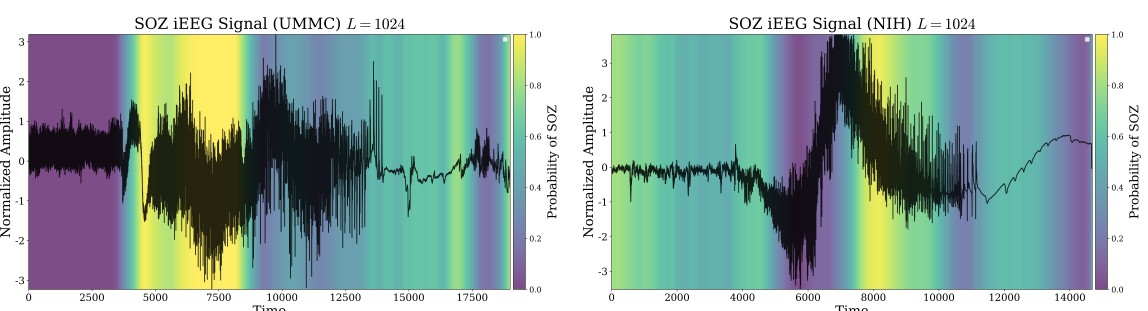

Figure 2: Visualization of SSS window probabilities throughout iEEG channels at inference time, using the PatchTST backbone with window size $1024$. The heatmap represents locally averaged window probabilities over time, with color intensity being proportional to the likelihood of the channel belonging to the SOZ.

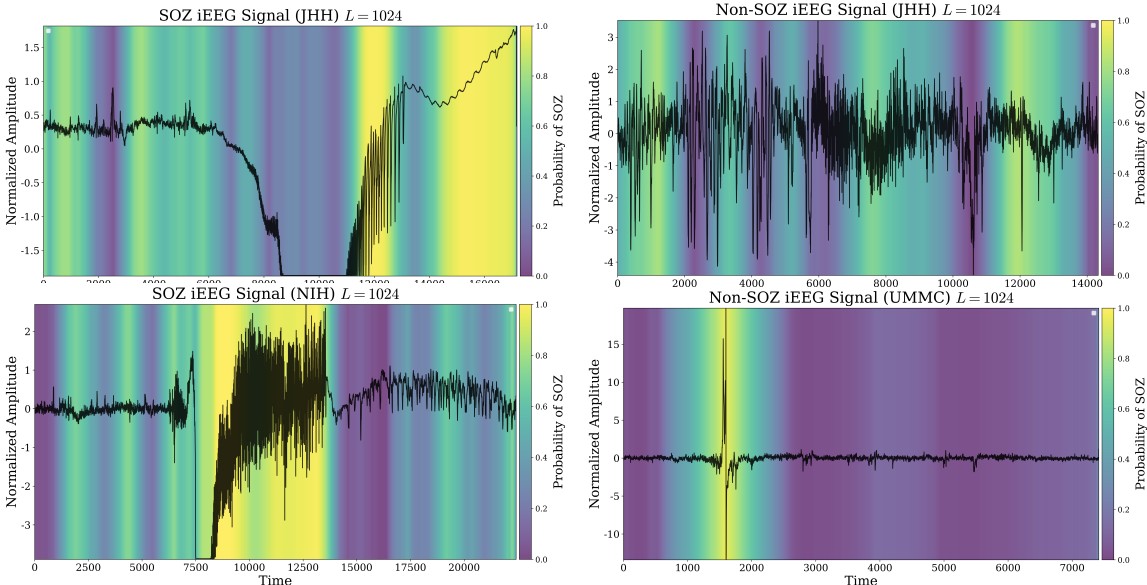

Figure 3: Visualization of SSS window probabilities for OOD iEEG channels at inference time, using the PatchTST backbone with window size $1024$. The heatmap represents locally averaged window probabilities over time, with color intensity being proportional to the likelihood of the channel belonging to the SOZ.

## 5    DISCUSSION

### 5.1    REVIEW OF RESULTS

In general, we observe that SSS outperforms modern finite-context and infinite-context methods both in-distribution and OOD for univariate VTSC. SOZ localization presents a significant challenge due to intra-patient and inter-patient variability, and with our evaluation the collection of 3 heterogeneous datasets, serving rigorous testbed for assessing the generalizability of SSS's capabilities for learning local signal characteristics in medical time series. While our experiments focus on univariate VTSC, as outlined in section 3.2.1 and 3.2.2, SSS can be easily applied to the multivariate setting.

Our results from the multicenter evaluation Table 1 indicate that SSS benefits from a diversity of data distributions and volume of training examples, given by the magnitude in performance differences, when compared to single cluster results. Furthermore, our OOD experiments from Table 2 suggest that SSS may be learning local signal characteristics present in different patient populations, leading to a advantage over finite-context and infinite-context methods. Our visualizations of SSS's predictions OOD in Figure 3 supports this notion, as there exist clear qualitative differences in locally averaged window probabilities with respect to anomalous signal characteristics, such as spikes or increases in amplitude or frequency. Figure 2 shows SSS's predictions in-distribution which also suggest a form of implicit semantic segmentation for anomalous local regions of the signal with respect to the SOZ probability. More analysis is needed to further solidify our understanding, which would benefit from a rigorous explainability study in future works.

### 5.2    CONCLUSION

To conclude, this work introduces novel VTSC framework, Stochastic Sparse Sampling (SSS), specifically tailored for medical time series applications. SSS blends the best of both worlds between finite-context methods (enabling usage of finite-context backbones) while allowing sampling of the entire signal in a computationally efficient manner that is less prone to context overload from infinite-context methods. SSS learns local signal characteristics, which provides the added benefit of inherent interpretability, and provides superior performance to the SOTA in-distribuion and OOD for unseen medical centers. For future work, it would be valuable to: (1) benchmark SSS across a wider variety of variable-length medical time series, (2) provide further rigorous post-hoc insights into the window probability distribution given by SSS, and (3) potentially include uncertainty estimates within the aggregation function to highlight anomalous regions of the signal.

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

# A  STOCHASTIC SPARSE SAMPLING

## A.1  SAMPLING IMPLEMENTATION

Let $X = \left\{ (\mathbf{x}_t^{(1)})_{t=1}^{T_1}, \ldots, (\mathbf{x}_t^{(n)})_{t=1}^{T_n} \right\}$ be the collection of variable-length time series. To achieve the sampling procedure outlined in section 3.2, we first construct the set all windows $\mathcal{W}$ by performing the slicing window method over each individual time series in $X$. For a window size $L$, with window stride $S$, and a time series $i$ with sequence length $T_i$, we obtain $A_i = \lfloor \frac{T_i - L}{S} \rfloor + 1$ windows. Note that $A_i \propto T_i$ for each $i$. During training, we convert $\mathcal{W}$ to a PyTorch dataset and use the native PyTorch dataloader with batch size $B$. When a batch is sampled, windows are drawn uniformly from $\mathcal{W}$, and thus for each $i$, the probability of observing a window from series $i$ is $p_i = A_i / (\sum_{j=1}^{n} A_j) \approx T_i / (\sum_{j=1}^{n} T_j)$. If $N_i$ represents the number of windows in the batch from series $i$, then we achieve the desired sampling property of $N_i \sim \text{Binomial}(B, p_i)$ where $p_i \approx T_i / (\sum_{j=1}^{n} T_j)$. Note that this procedure uses sampling *without* replacement; one may consider replacement, however, we did not experiment with this and leave modifications with more complex sampling procedures as a future direction.

## A.2  ABLATIONS

### A.2.1  BATCH SIZE

Table 3 reports mean F1 score, AUC, and accuracy (%) with standard deviations, are reported over 5 seeds, for the evaluation on all medical centers. Each experiment uses the best configuration described in Table 6. While the aggregation function in Equation (3) may benefit from a higher number of samples within the batch (due to mean approximation), we observe that the performance of SSS remains relatively constant across various batch sizes, and thus does not require large batch sizes to achieve adequate performance.

Table 3: Performance of SSS over various batch sizes.

| Batch Size | F1 Score | AUC | Acc. (%) |
|---|---|---|---|
| 128 | $0.7563 \pm 0.02717$ | $0.8139 \pm 0.04302$ | $70.79 \pm 2.323$ |
| 512 | $0.7498 \pm 0.02354$ | $0.7965 \pm 0.03526$ | $69.46 \pm 3.109$ |
| 2048 | $0.7651 \pm 0.03568$ | $0.8194 \pm 0.05166$ | $71.40 \pm 6.078$ |
| 4096 | $0.7441 \pm 0.02987$ | $0.7979 \pm 0.06818$ | $68.09 \pm 4.848$ |
| 8192 | $0.7629 \pm 0.02829$ | $0.7999 \pm 0.05331$ | $72.35 \pm 4.965$ |

### A.2.2  WINDOW SIZE

Table 4 follows the same experimental setup as Table 3, but varies over the window size $L$ instead of batch size. While the performance of $L = 512$ and $L = 1024$ remain relatively on par, we notice that the F1 score drops significantly for $L = 2048$ along with all other metrics. This suggests that a large receptive field may not be advantageous, and that SSS benefits from processing localized areas of the signal. Indeed, as $L$ increases we expect to reach a similar performance to the finite-context PatchTST baseline, with a decrease in performance as a result.

Table 4: Performance of SSS over various window sizes $L$.

| $L$ | F1 Score | AUC | Acc. (%) |
|---|---|---|---|
| 512 | $0.7567 \pm 0.01075$ | $0.8141 \pm 0.03054$ | $70.62 \pm 2.165$ |
| 1024 | $0.7629 \pm 0.02829$ | $0.7999 \pm 0.05331$ | $72.35 \pm 4.965$ |
| 2048 | $0.7334 \pm 0.03003$ | $0.7719 \pm 0.04762$ | $68.52 \pm 3.353$ |

### A.2.3  CALIBRATION

Table 4 follows the same experimental setup as Table 3, but varies over the window size $L$ instead of batch size. While the performance of $L = 512$ and $L = 1024$ remain relatively on par, we notice that the F1 score drops significantly for $L = 2048$ along with all other metrics. This suggests that a large receptive field

may not be advantageous, and that SSS benefits from processing localized areas of the signal. Indeed, as $L$ increases we expect to reach a similar performance to the finite-context PatchTST baseline, with a decrease in performance as a result.

## A.3 CONVEX AGGREGATION

**Theorem 1** (Probability Distribution Guarantee). *Fix $K, n \in \mathbb{N}$. Suppose $\alpha_1, \ldots, \alpha_n \geq 0$ satisfies $\sum_{i=1}^{n} \alpha_i = 1$, and $\mathbf{v}_1, \ldots, \mathbf{v}_n \in [0,1]^K$ each satisfy $\sum_{j=1}^{K} v_{ik} = 1$ for $1 \leq i \leq n$. That is, each $\mathbf{v}_i = (v_{i1}, v_{i2}, \ldots, v_{iK})^T$ represents a valid discrete probability distribution over $K$ classes. Then the convex combination:*

$$\mathbf{y} = \sum_{i=1}^{n} \alpha_i \mathbf{v}_i, \tag{5}$$

*also represents a valid discrete probability distribution, satisfying $\sum_{j=1}^{K} y_j = 1$.*

*Proof.* By construction, $y_j = \sum_{i=1}^{n} \alpha_i v_{ij}$ for each entry $1 \leq j \leq K$. Then $y_j \geq 0$, since for each $1 \leq i \leq n$ and $1 \leq j \leq K$ we are given that $\alpha_i \geq 0$ and $v_{ij} \geq 0$. Furthermore, we can write:

$$\sum_{j=1}^{n} y_j = \sum_{j=1}^{K} \sum_{i=1}^{n} \alpha_i v_{ij}$$

$$= \sum_{i=1}^{n} \sum_{j=1}^{K} \alpha_i v_{ij} \qquad \text{(Swap summation order)}$$

$$= \sum_{i=1}^{n} \alpha_i \sum_{j=1}^{K} v_{ij}$$

$$= \sum_{i=1}^{n} \alpha_i \qquad (\textstyle\sum_{j=1}^{K} v_{ij} = 1 \text{ for all } i)$$

$$= 1 \qquad (\textstyle\sum_{i=1}^{n} \alpha_i = 1)$$

It follows that since each $y_j \geq 0$ and $\sum_{j=1}^{K} y_j = 1$, then $\mathbf{y}$ represents a valid discrete probability distribution over $K$ classes. $\square$

## A.4 CALIBRATION

Let $\hat{y}_1, \ldots, \hat{y}_n \in [0, 1]$ be the uncalibrated window probabilities, each with a corresponding binary label $y_1, \ldots, y_n \in \{0, 1\}$ derived from the label of the time series; that is, if $\hat{y}_i = f_\theta(\mathbf{w}_i)$ for a window $\mathbf{w}_i$, then the window label $y_i$ is inherited from the global time series it was sampled from. The goal of probability calibration is to transform each $\hat{y}_i$ into $\tilde{y}_i = g_\phi(\hat{y}_i)$, such that $\tilde{y}_i$ represents true likelihood of a class. Within this context, probability calibration can help mitigate the impact of temporal fluctuations and local anomalies by adjusting probabilities for each individual windows. Note that while

Table 5: Performance of SSS with different calibration methods.

| Calibration Method | F1 Score | AUC | Acc. (%) |
|---|---|---|---|
| Isotonic Regression | $0.7629 \pm 0.02829$ | $0.7999 \pm 0.05331$ | $72.35 \pm 4.965$ |
| Venn-ABERS | $0.7637 \pm 0.02704$ | $0.8003 \pm 0.05308$ | $72.47 \pm 4.773$ |
| No Calibration | $0.7291 \pm 0.06909$ | $0.7830 \pm 0.03844$ | $69.93 \pm 4.635$ |

calibration may yield more refined probability estimates with reduced noise, it is an integral intermediate step rather than a global optimization procedure on the final probabilities. For each calibration we considered, we provide a short description below.

**Isotonic regression** is a nonparametric method that fits a weighted least-squares model subject to motonicity constraints (Silvapulle & Sen, 2011). Formally, this can be stated as a quadratic program (QP) given by:

$$\min_g \sum_{i=1}^{n} w_i(\tilde{y}_i - y_i)^2 \text{ subject to } \tilde{y}_i \leq \tilde{y}_j \text{ for all } i, j \text{ where } \hat{y}_i \leq \hat{y}_j. \tag{6}$$

where $\tilde{y}_i = g(\hat{y}_i)$ and $w_i \geq 0$ are weights assigned to each datapoint, which are often each set to $w_i = 1$ to provide equal importance over all inputs. The monotonicity constraint ensures that uncalibrated probabilities will always map to equal or higher calibrated probabilities. Due its nonparametric nature, isotonic regression can adapt to various probability distributions across diverse datasets. However, this flexibility comes at a cost as it is also prone to overfitting on smaller datasets, potentially adjusting to noise rather than properly calibrating its inputs.

**Venn-Abers predictors** is based on the concept of isotonic regression but extends it to ensure validity within the framework of conformal prediction, which provides uncertainty estimates with distribution-free theoretical guarantees (Vovk & Petej, 2014; Angelopoulos & Bates, 2021). For an uncalibrated probability $\hat{y}$, two isotonic calibrators are trained:

$$p_0 = g_0(\hat{y}) \text{ and } p_1 = g_1(\hat{y}) \tag{7}$$

where $g_0$ and $g_1$ are isotonic functions derived from augmented sets. These sets include $(\hat{y}, 0)$ and $(\hat{y}, 1)$ respectively, alongside all other uncalibrated probabilities and their respective labels. The values $p_0$ and $p_1$ represent likelihoods for class 0 and class 1, while the interval $[p_0, p_1]$ provides an uncertainty estimate of where the true probability resides. The final calibrated probability is then given by:

$$\tilde{y} = \frac{p_1}{1 - p_0 + p_1}. \tag{8}$$

Venn-Abers predictors provide guaranteed validity in terms of calibration, meaning the predicted probabilities closely match empirical frequencies. While we do not explicitly utilize the uncertainty interval $[p_0, p_1]$ (only the calibrated score $\tilde{y}$), this method can be effective in scenarios requiring risk assessment or critical tasks where rigorous uncertainty estimates are crucial. We leave this as a future direction to implement conformal prediction within the context of SSS, to provide uncertainty guarantees based off of window predictions, which may be useful for post-hoc interpretability. In comparsion to isotonic regression, Venn-Abers can be more computationally intensive, as it requires fitting two isotonic functions simultaneously.

## A.5 FINITE-CONTEXT & INFINITE-CONTEXT METHODS

**Definition 2.** Let $\mathcal{X}$ be a vector space over $\mathbb{R}$ and $f_\theta : \mathcal{X} \to \mathcal{Y}$ be a model with parameters $\theta$ and output space $\mathcal{Y}$. We say that $f_\theta$ has **finite-context** if $\mathcal{X}$ is finite-dimensional, that is, there exists some $n \in \mathbb{N}$ such that $\mathcal{X} \cong \mathbb{R}^n$ as vector spaces. Whereas $f_\theta$ is said to have **infinite-context** if $\mathcal{X} = \mathbb{R}^{(\infty)}$ is the space of real number sequences with finite support[2].

Note that this definition refers to the *native* capabilities of $f_\theta$, without the usage of data manipulation techniques such as padding, truncation, and interpolation. We utilize this formalization to separate our baselines, so that we may better understand the advantages and limitations of both.

Table 6: Hyperparameter search space for SSS (with PatchTST backbone). Best configuration is highlighted in red.

| Parameter | Search Values |
|---|---|
| $d_{\text{model}}$ | $\{16, 32, 64\}$ |
| $d_{\text{ff}}$ | $\{32, 64, 128\}$ |
| num_heads | $\{2, 4, 8\}$ |
| num_enc_layers | $\{1, 2, 3\}$ |
| lr | $\{10^{-4}, 10^{-5}\}$ |
| $L$ | $\{512, 1024, 2048\}$ |
| batch_size | $\{2048, 4096, 8192\}$ |
| $g_\phi$ | $\{\text{isotonic regression}, \text{Venn-Abers predictors}\}$ |

### A.6 SSS IMPLEMENTATION AND CONFIGURATIONS

## B DATASET AND PREPROCESSING

### B.1 DATASET

Table 7: iEEG Multicenter Dataset Summary: For each medical center, we report the total number of patients recorded ($n$), the number of patients with seizure onset zone (SOZ) annotations ($n_{\text{SOZ}}$), the number of time series recordings ($n_{\text{ts}}$), the percentage of time series labeled as SOZ ($p_{\text{SOZ}}$), the type of iEEG method used (e.g., electrocorticography, ECoG), the sampling frequency (Hz) (noting that some recordings may vary), and post-operative patient outcomes following SOZ surgical resection.

| Medical Center | $n$ | $n_{\text{SOZ}}$ | $n_{\text{ts}}$ | $p_{\text{SOZ}}$ | iEEG Type | Frequency (Hz) | Patient Outcomes |
|---|---|---|---|---|---|---|---|
| JHH | 7 | 3 | 1458 | 7.48% | ECoG | 1000 | No |
| NIH | 14 | 11 | 3057 | 12.23% | ECoG | 1000 | Yes |
| UMMC | 9 | 9 | 2967 | 5.56% | ECoG | 250-1000 | Yes |
| UMF | 5 | 1 | 129 | 25.58% | ECoG | 1000 | No |

Table 7 provides an overview of the iEEG Multicenter Dataset. For each cluster, we filter out patients ($n$) who have SOZ annotations ($n_{\text{SOZ}}$). All channels for all patients are group together into one dataset per medical center, where $n_{\text{ts}}$ indicates the number of examples. However, due to the heavy imbalance between SOZ-labeled time series and non-SOZ labeled time series, the number of examples used for training and validation decreases significantly once we employ class balancing, resulting in $\lfloor 2 \cdot p_{\text{SOZ}} \cdot n_{\text{ts}} \rfloor$ examples for each medical center, which is then split into training, validation, and testing.

### B.2 DATA PREPROCESSING

Each patient recording contains multiple channels corresponding to individual electrodes from the iEEG device. During preprocessing, for all patients, we extract all channels and balance the dataset to have an equal number of SOZ and non-SOZ channels. After, we partition this dataset into train, validation, and test channels with a 70%/10%/20% split respectively, and ensure that during the window sampling phase of SSS there is no temporal leakage from the test set. Each channel, or univariate time series, is $z-$score normalized to have zero mean and unit standard deviation.

---

[2]Every sequence in $\mathbb{R}^{(\infty)}$ must have finitely many non-zero terms.

Due to the extremely long sequence length of several channels, we required downsampling to fit the dataset into memory (250GB RAM). To achieve this, for each channel we applied a 1D average pooling layer with `kernel_size=24` and `kernel_stride=12` before feeding it to the baseline model or before performing the window sampling procedure for SSS at train-time.

Finite-context methods required either padding, truncation, or interpolation due to fit each sequence into its limited context window. For each finite-context method we perform a combination of padding and truncation according to the chosen window size $L$: if the sequence length of the original time series exceeded $L$, it was truncated to obtain the first $L$ time points, otherwise if it was less than $L$, it was padded with zeros at the end of the sequence to ensure a sequence length of $L$.

### B.3 EVALUATION METRICS

#### B.3.1 F1

The F1 score is defined as the harmonic mean of precision and recall, given by:

$$F_1 = 2 \cdot \frac{\text{Precision} \cdot \text{Recall}}{\text{Precision} + \text{Recall}}, \tag{9}$$

where,

$$\text{Precision} = \frac{\text{True Positives (TP)}}{\text{True Positives (TP)} + \text{False Positives (FP)}}, \tag{10}$$

and,

$$\text{Recall} = \frac{\text{True Positives (TP)}}{\text{True Positives (TP)} + \text{False Negatives (FN)}}. \tag{11}$$

We select the F1 score as our primary evaluation metric for SOZ localization for the following reasons. The F1 score balances the need to correctly identify all regions of the SOZ (recall) with the need to avoid misclassifying healthy regions as SOZ (precision). This balance is crucial in surgical planning, where both missing SOZs and unnecessarily removing healthy tissue can have severe consequences. Unlike accuracy, which overlooks the difference between false positives and false negatives, the F1 score provides a more nuanced evaluation by considering both, making it well-suited in clinical contexts such as SOZ localization.

#### B.3.2 AUC

We complement the F1 score with the Area Under the Receiver Operating Characteristic curve (AUC), defined by:

$$\text{AUC} = \int_0^1 \text{TPR}(t) \cdot \frac{d\text{FPR}(t)}{dt} dt \tag{12}$$

While the F1 score provides insight into the balance between precision and recall at a specific threshold, AUC assesses the model's overall discriminative ability across all thresholds. This threshold-independent evaluation is relevant for critical scenarios where the threshold maybe be adjusted from $0.5$, which is not common in clinical settings.

## C IMPLEMENTATION AND EXPERIMENTAL CONFIGURATIONS

For each baseline, we perform grid search and optimize with respect to best accuracy score on the evaluation for all medical centers. $L$ refers to the window size parameter, $d_{\text{model}}$ is the model dimension, and $d_{\text{ff}}$

is the dimension of the feed-forward network. The grid search parameters for each baseline are shown below; for information on the implementation of SSS, see Appendix A.6. In all experiments, we train using the Adam optimizer (Kingma & Ba, 2014), for 50 epochs, with cosine learning rate annealing (one cycle with 50 epochs in length) which adjusts the learning rate down by two orders of magnitude (e.g., $10^{-4}$ to $10^{-6}$) by the last epoch. We also implement early stopping with a patience of 15, and apply learnable instance normalization (Kim et al., 2021) for each input. For most of the baselines we use a dropout rate of $0.2 - 0.3$, and weight decay to $10^{-4} - 10^{-5}$, but do not explicitly tune these parameters in our grid search. For finite-context methods we set the batch size to the entire dataset (596 individual univariate time series for all clusters), whereas infinite-context methods required batch size of 1 due to their variable-length. The code for SSS and baseline implementations is available at the following anonymous link: https://anonymous.4open.science/r/sss-0D75/.

**PatchTST**: We adapt the official implementation github.com/yuqinie98/PatchTST, but swap out the attention module with the native PyTorch torch.nn.MultiheadAttention module.

Table 8: Hyperparameter search space for PatchTST. Best configuration is highlighted in red.

| Parameter | Search Values |
| --- | --- |
| $d_{\mathrm{model}}$ | $\{16, 32, 64\}$ |
| $d_{\mathrm{ff}}$ | $\{32, 64, 128\}$ |
| num_heads | $\{2, 4, 8\}$ |
| num_enc_layers | $\{1, 2, 3\}$ |
| lr | $\{10^{-4}, 10^{-5}\}$ |
| $L$ | $\{1000, 3000, 5000, 10000\}$ |

**TimesNet**: We use the official implementation github.com/thuml/Time-Series-Library.

Table 9: Hyperparameter search space for TimesNet. Best configuration is highlighted in red.

| Parameter | Search Values |
| --- | --- |
| $d_{\mathrm{model}}$ | $\{16, 32, 64\}$ |
| $d_{\mathrm{ff}}$ | $\{32, 64, 128\}$ |
| num_kernels | $\{4, 6\}$ |
| top_k | $\{3, 5\}$ |
| num_enc_layers | $\{1, 2\}$ |
| lr | $\{10^{-4}, 10^{-5}\}$ |
| $L$ | $\{1000, 3000, 5000, 10000\}$ |

**ModernTCN**: We use the official implementation `github.com/luodhhh/ModernTCN`.

Table 10: Hyperparameter search space for ModernTCN. Best configuration is highlighted in red.

| Parameter | Search Values |
|---|---|
| lr | $\{10^{-4}, 10^{-5}\}$ |
| $d_{\text{model}}$ | $\{16, 32, 64\}$ |
| num_enc_layers | $\{1, 2\}$ |
| large_size_kernel | $\{9, 13, 21, 51\}$ |
| small_size_kernel | $5$ |
| dw_dims | $\{128, 256\}$ |
| ffn_ratio | $\{1, 4\}$ |
| $L$ | $\{1000, 3000, 5000, 10000\}$ |

**DLinear**: We use the implementation from `github.com/thuml/Time-Series-Library`.

Table 11: Hyperparameter search space for DLinear. Best configuration is highlighted in red.

| Parameter | Search Values |
|---|---|
| moving_avg | $\{10, 25\}$ |
| lr | $\{10^{-4}, 10^{-5}, 10^{-6}\}$ |
| $L$ | $\{1000, 3000, 5000, 10000\}$ |

**ROCKET**: We use the official implementation `github.com/angus924/rocket` and follow the standard implementation of $10,000$ kernels.

Table 12: Hyperparameter search space for ROCKET. Best configuration is highlighted in red.

| Parameter | Search Values |
|---|---|
| num_kernels | $\{10000\}$ |
| lr | $\{10^{-4}, 10^{-5}, 10^{-6}\}$ |

**Mamba**: We use the package `mambapy` which builds upon the official Mamba implementation. We

Table 13: Hyperparameter search space for Mamba. Best configuration is highlighted in red.

| Parameter | Search Values |
|---|---|
| lr | $\{10^{-4}, 10^{-5}, 10^{-6}\}$ |
| $d_{\text{model}}$ | $\{16, 32, 64\}$ |
| num_enc_layers | $\{1, 2, 3\}$ |

also employ patching from (Nie et al., 2023), which we observed led to greater to performance, with $\text{patch\_size} = 64$ and $\text{patch\_stride} = 16$.

**GRUs**: We utilized the native PyTorch module `torch.nn.GRU`.

Table 14: Hyperparameter search space for GRUs. Best configuration is highlighted in red.

| Parameter | Search Values |
|---|---|
| lr | $\{10^{-4}, 10^{-5}, 10^{-6}\}$ |
| $d_{\text{model}}$ | $\{16, 32, 64\}$ |
| num_enc_layers | $\{1, 2, 3\}$ |
| bidirectional | $\{\text{True}, \text{False}\}$ |

## C.1 COMPUTATIONAL RESOURCES

Our experiments were conducted on 4x NVIDIA RTX 6000 Ada Generation GPUs using the PyTorch framework with CUDA Paszke et al. (2019). Although we have not tracked explicitly the amount of consumed GPU hours, all experiments can be conducted for 5 seeds no more than $2 - 3$ hours with a similar setup.