# OpenReview forum: "Stochastic Sparse Sampling: A Framework for Variable-Length Medical Time Series Classification"
_ICLR.cc/2025/Conference — ICLR 2025 Conference Withdrawn Submission_

### Official Review · Reviewer_7ats · 2024-11-02

**Soundness:** 2
**Presentation:** 2
**Contribution:** 2
**Rating:** 5
**Confidence:** 5

**Summary:**

This paper introduces a multi-scale learning approach for medical time series classification. The proposed method comprises multiple independent models, each with a distinct patch length, allowing it to capture information across various temporal scales. The patching method follows the PatchTST framework, which employs single-channel patching. To reduce computational costs, the authors implement stochastic sparse sampling, randomly selecting models during training. The final representation is an aggregation of outputs from all models, combining multi-scale information. The model is evaluated on intracranial EEG (iEEG) data for seizure onset zone classification, using a dataset collected from four independent medical centers.

**Strengths:**

The use of sparse sampling for computational savings in multi-scale learning is an interesting idea. Additionally, the out-of-distribution classification on unseen subjects from different medical centers demonstrates strong potential for generalizability in real-world applications.

**Weaknesses:**

The motivation to save computational resources is well-intentioned, though I am concerned about its practicality in actual training. For a given set of window sizes with corresponding independent models, even if only subsets of window sizes are selected during training, the space complexity of the models remains unchanged. This approach primarily improves training speed without reducing memory requirements. Additionally, using an independent model for each patch length may not be optimal for memory efficiency. A shared backbone across different patch lengths could be a more effective choice for memory savings. Overall, the method resembles an enhanced version of MTST [1], employing a random subset of models with varying patch lengths during training.

Moreover, while the paper’s title refers to medical time series, only a single seizure dataset is used for evaluation. Expanding the evaluation to include additional datasets would strengthen the claim of generalizability. The ablation study could also benefit from a deeper investigation into multi-scale learning with various patch lengths. For instance, exploring which combinations of patch lengths yield the best performance would be informative. Additionally, the impact of stochastic sparse sampling should be assessed in detail. For a given list of patch lengths, how do memory usage and running time between training with and without stochastic sparse sampling? Lastly, a recent work, Medformer[2], should be compared in baseline methods, as it is also designed for medical time series classification using multi-scale patching. A discussion on the differences between this method and Medformer would also be valuable for highlighting the unique aspects of the proposed approach.


[1] Multi-resolution Time-Series Transformer for Long-term Forecasting
[2] Medformer: A Multi-Granularity Patching Transformer for Medical Time-Series Classification

**Questions:**

See Weaknesses

---

### Official Review · Reviewer_xZmY · 2024-11-02

**Soundness:** 2
**Presentation:** 2
**Contribution:** 1
**Rating:** 3
**Confidence:** 4

**Summary:**

This paper introduces Stochastic Sparse Sampling (SSS), a new framework for classifying variable-length medical time series. SSS employs fixed windows sparsely to make local predictions, which are then combined to form a global prediction.

**Strengths:**

1) Seizure onset zone (SOZ) detection is a novel direction that may be important for clinical intracranial EEG (iEEG) research.
2) The method is described in detail, ensuring clarity and reproducibility.
3) The experiments use a large-scale public iEEG dataset for evaluation, showing the robustness and effectiveness across diverse and heterogeneous data sources.

**Weaknesses:**

1) This paper uses a single-channel approach for SOZ detection, justifying it by citing the challenges posed by varying numbers of iEEG channels across iEEG recordings. However, this reason does not demonstrate that single-channel analysis is more effective. In clinical practice, SOZ and early propagation zones involve multiple channels with timing differences in abnormal discharges [1]. The authors do not provide sufficient theoretical or experimental support to explain why a single-channel approach would effectively capture these critical distinctions.
2) This paper defines SOZ detection as a variable-length time series classification (VTSC) task. Still, the authors do not clarify the specific benefits of VTSC over general anomaly detection for this application. Their justification, that “effective treatment requires analysis of variable-length signals,” lacks concrete references or explanations relevant to variable-length signals, weakening the rationale for using VTSC in this context.
3) The proposed method is quite simple. How does it differ from other downsampling or sparse sampling approaches?  Model output still provides a global classification result, which seems no different from a standard classification model. The paper should compare its approach with more methods that share a similar motivation and more details of related works to clarify its position within the research field.
4) The authors state in the abstract that their method outperforms "state-of-the-art (SOTA) baselines across most medical centers." However, none of these baselines were designed for the SOZ detection task or medical time series data.Instead, all baseline models were built for general time series or other sequential data forecasting tasks, not even for time series classification. The authors should have used baselines developed for iEEG or EEG signal analysis, such as methods in references [2-4], or, at the very least, models designed for time series classification or anomaly detection [4,5].


[1] Li et al., Neural fragility as an EEG marker of the seizure onset zone, Nature Neuroscience, 2021.

[2] Tang et al., Self-Supervised Graph Neural Networks for Improved Electroencephalographic Seizure Analysis, ICLR, 2022.

[3] Luo et al., Exploring Adaptive Graph Topologies and Temporal Graph Networks for EEG-Based Depression Detection, IEEE Transactions on Neural System and Rehabilitation Engineering, 2023.

[4] Rikuto et al., SplitSEE: A Splittable Self-supervised Framework for Single-Channel EEG Representation Learning, ICDM, 2024.

[5] tang et al., Omni-Scale CNNs: a simple and effective kernel size configuration for time series classification, ICLR, 2022.

[6] Lu et al., Out-of-Distribution Representation Learning for Time Series Classification, ICLR, 2023.

**Questions:**

My main concern is the motivation. Clinically, the brain regions requiring surgical removal include the SOZ and early propagation zones. These areas contain numerous neurons that exhibit abnormal discharges and are typically distributed across multiple iEEG channels. These regions show the earliest electrophysiological changes in a seizure, often before clinical symptoms appear.
Suppose the model can only consider a single iEEG data channel. In that case, it may miss the timing differences of abnormal discharges across channels, making it difficult to determine which neurons initiated the seizure.
Experts have developed many graph topology-based methods and biomarkers to capture these distinctions.
In this paper, the authors justify their choice of a single-channel approach by explaining that varying numbers of iEEG channels across recordings create challenges for developing multi-channel analysis methods.
However, this limitation alone does not prove that single-channel analysis is more effective. Could the authors offer more theoretical or experimental support to justify the potential effectiveness of a single-channel method?

---

### Official Review · Reviewer_MvPo · 2024-11-02

**Soundness:** 3
**Presentation:** 3
**Contribution:** 3
**Rating:** 6
**Confidence:** 4

**Summary:**

The paper describes using aggregation of time-series classification model predictions across windows during training and inference a way to go beyond fixed-context length window processing and infinite context models recurrent neural networks.  The aggregation method explored here is simple averaging. An additional calibration step is used after the model is trained.

The method is applied to different EEG channels in order to learn to classify a channel as being in the seizure originating zone or not. Cross-subject and cross institution results show very promising performance compared to fixed-context length approach and infinite context models.

**Strengths:**

Very clear presentation and well-fit for this type of time series classification problem. The calibration step after pooling during training is a thoughtful addition.

**Weaknesses:**

Main concern is the single domain/task used to test the method. While the single domain is very interesting, there is something different in the fact that the seizure periods are themselves randomly occurring throughout the time series. In other tasks, long-term dynamics of the time series may require extracting patterns through time rather than this which is more akin to multiple instance learning where the search is for any evidence of positive class.

The lack of other tasks weakens the generality of the method, but I don't have the perfect case of variable length it is hard to say where there would

**Questions:**

In the main body, the discussion of what portion of the training or validation set is used for the calibration is missing.

Line 797 "with respect to best accuracy score on the evaluation" . This doesn't seem like a valid hyper-parameter selection if this is done on test instead of validation. Also it should be noted that hyper-parameter is selected in terms of accuracy, even if F1 and AUC are reported.

In Algorithm 1, the windows from the time series are sampled proportional to their length. It would seem that stratified sampling by label or group may be motivated in cases of imbalanced training data.  This could be mentioned.

Typos:
180 "by sampling with replacement," -> "by sampling without replacement,"

268 "the the correct "

388 "a advantage"

---

### Official Review · Reviewer_mZBK · 2024-11-04

**Soundness:** 2
**Presentation:** 3
**Contribution:** 3
**Rating:** 5
**Confidence:** 3

**Summary:**

This article proposes a meta-heuristic to improve time series classification algorithms by classifying random windows and aggregating scores. The algorithm is tested on a real-world task where it achieves promising performance.

**Strengths:**

- The article is well-written and easy to follow.
- The methodology is sound and has the potential to provide an interpretable time series classification strategy that can be combined with virtually any time series classification algorithm.

**Weaknesses:**

- The proposed approach is simple, which is perfectly fine, but lacks a more thorough analysis, at least empirical. For a reader with a use case in mind, it is difficult to assess if this method is appropriate.
- Another drawback is that this approach has only been tested on one data set. To assess its generalization capability, the authors could test their method on additional publicly available time series datasets.
- Section 4.5 on qualitative interpretation needs to (re)written to be more convincing as it lacks much information or comments. Interpretability is one of the contributions listed in the introduction.

**Questions:**

- Approaches based on convolutions can handle variable lengths, and the number of parameters does not increase with the signal length, contrary to what is stated in the section on related work. Can the authors clarify or revise their statement about convolutional approaches in the related work section?
- The sampling is said to be sparse. How does the performance vary with the number of drawn windows? Consider doing an ablation study or experiment showing performance as a function of the number of sampled windows.
- To improve the statistical rigor of the results, can the authors report standard deviations or confidence intervals for all reported performance metrics?
- Certain notations can be misleading : $y^{(i)}$ is an integer while $\hat{y}^{(i)}$ is vector. Consider changing them to more consistent notations.
- Current baselines are mostly deep-learning-based, but many other classification algorithms exist [1]. Can the authors include a few other baselines that they think are relevant?
- In Section 4.5, what are the labels of the displayed signals? Do they belong to the same individual? Are the highlighted areas related to a known physiological phenomenon?
- How important is the smoothing step presented in Section 3.2.2? Its influence could be for instance illustrated with an ablatio study.

[1] Ruiz, A. P., Flynn, M., Large, J., Middlehurst, M., & Bagnall, A. (2021). The great multivariate time series classification bake off: a review and experimental evaluation of recent algorithmic advances. Data Mining and Knowledge Discovery, 35(2).

---

### Note · Authors · 2024-11-24

**Comment:**

We appreciate the reviewers' time and detailed feedback, which will be valuable for improving our work, but we have decided to withdraw this paper from consideration at the conference. Thank you again for your thoughtful comments.

**Withdrawal Confirmation:**

I have read and agree with the venue's withdrawal policy on behalf of myself and my co-authors.